# Exploring Transfer Learning in Medical Image Segmentation using Vision-Language Models

**Kanchan Poudel**\*               KANCHAN.POUDEL@NAAMII.ORG.NP
**Manish Dhakal**\*                MANISH.DHAKAL@NAAMII.ORG.NP
**Prasiddha Bhandari**\*            PRASIDDHA.BHANDARI@NAAMII.ORG.NP
**Rabin Adhikari**\*               RABIN.ADHIKARI@NAAMII.ORG.NP
**Safal Thapaliya**\*              SAFAL.THAPALIYA@NAAMII.ORG.NP
**Bishesh Khanal**                BISHESH.KHANAL@NAAMII.ORG.NP
*Nepal Applied Mathematics and Informatics Institute for research (NAAMII), Nepal*

**Editors:** Accepted for publication at MIDL 2024

## Abstract

Medical image segmentation allows quantifying target structure size and shape, aiding in disease diagnosis, prognosis, surgery planning, and comprehension. Building upon recent advancements in foundation Vision-Language Models (VLMs) from natural image-text pairs, several studies have proposed adapting them to Vision-Language Segmentation Models (VLSMs) that allow using language text as an additional input to segmentation models. Introducing auxiliary information via text with human-in-the-loop prompting during inference opens up unique opportunities, such as open vocabulary segmentation and potentially more robust segmentation models against out-of-distribution data.

Although transfer learning from natural to medical images has been explored for image-only segmentation models, the joint representation of vision-language in segmentation problems remains underexplored. This study introduces the first systematic study on transferring VLSMs to 2D medical images, using carefully curated 11 datasets encompassing diverse modalities and insightful language prompts and experiments. Our findings demonstrate that although VLSMs show competitive performance compared to image-only models for segmentation after finetuning in limited medical image datasets, not all VLSMs utilize the additional information from language prompts, with image features playing a dominant role. While VLSMs exhibit enhanced performance in handling pooled datasets with diverse modalities and show potential robustness to domain shifts compared to conventional segmentation models, our results suggest that novel approaches are required to enable VLSMs to leverage the various auxiliary information available through language prompts. The code and datasets are available at https://github.com/naamiinepal/medvlsm.

## 1. Introduction

Medical image segmentation is crucial for various clinical applications such as diagnosis, prognosis, and surgery planning. The latest supervised segmentation models exhibit promising outcomes across diverse imaging modalities, anatomies, and diseases (Milletari et al., 2016; Havaei et al., 2017; Zhou et al., 2018; Chen et al., 2021; Isensee et al., 2021; Hatamizadeh et al., 2022; Oktay et al., 2022; Wazir and Fraz, 2022). Despite their success, these models are constrained to predefined foreground classes on specific modalities and

---

\* Contributed equally. The order is in the ascending order of the authors' first names.

anatomies, lacking adaptability to auxiliary information and hindering their application outside extensive population-based studies.

The integration of VLMs (Huang et al., 2020; Jia et al., 2021; Li et al., 2021; Radford et al., 2021; Fürst et al., 2022; Singh et al., 2022; Zhai et al., 2022) into VLSMs (Lüddecke and Ecker, 2022; Rao et al., 2022; Wang et al., 2022) presents a paradigm shift in medical image segmentation. Models like CLIP (Radford et al., 2021) and BiomedCLIP (Zhang et al., 2023a), capable of joint text-image representation, allow for auxiliary information incorporation through language prompts during segmentation. This approach can enhance interpretability and robustness against domain shift and out-of-distribution data.

While transfer learning from natural to medical images for image-only representation learning has been extensively explored (Ghafoorian et al., 2017; Cheplygina et al., 2019; Amin et al., 2019), only a few such studies have been done for joint vision-language representation (Qin et al., 2022). Yet, two critical questions persist (**i**) the generalizability of this approach across multiple VLSMs for segmentation tasks, and (**ii**) the nuanced role of language prompts vs. images during finetuning and the VLSMs' capacity to handle pooled dataset training and out-of-distribution data.

This work presents the first systematic study on VLSM transfer learning to the medical images, using four models based on the two most popular contrastive VLMs: CLIP pretrained on natural image-text pairs and BiomedCLIP pretrained in the medical domain.

Key contributions include meticulous dataset selection (11 datasets) across four 2D medical image modalities, diverse anatomical structures, and pathology. We also enrich existing datasets with diverse language prompts generated through automated methods utilizing image metadata, VQA models, and segmentation masks. Our extensive experiments with four VLSMs, diverse datasets, and carefully designed prompts explore intricate relationships between language and image during joint representation adaptation for medical images. We evaluate robustness against domain shift and the ability to handle pooled datasets with diverse modalities, attributes, and targets. Finally, we open-source our framework, source code, and prompts, promoting transparency and reproducibility in the scientific community.

## 2. Method

### 2.1. CLIP- and BiomedCLIP-based Medical VLSMs

We create four medical VLSMs using CLIP and BiomedCLIP: (**i**) Finetuning CLIP-based VLSMs, **CLIPSeg** (Lüddecke and Ecker, 2022) and **CRIS**[1] (Wang et al., 2022), pretrained on natural image-text pairs, and (**ii**) Building two new VLSMs for the medical domain by adding a decoder to BiomedCLIP, pretrained on medical image-text pairs. The proposed new models are **BiomedCLIPSeg-D** (with a pretrained CLIPSeg decoder) and **Biomed-CLIPSeg** (with a randomly initialized decoder of CLIPSeg). A sample from the datasets in our experiments is a triplet of a medical image, a segmentation mask, and a text prompt. Figure 1 displays the overall VLSM architecture.

CLIPSeg accommodates both CNN and ViT (Dosovitskiy et al., 2020) backbones, whereas CRIS only supports a CNN-based CLIP backbone. BiomedCLIPSeg-based models include transformer-based backbones for both the encoders. We study CLIPSeg and CRIS

---

1. We used unofficial weights from a GitHub issue since the authors haven't released the model weights yet.

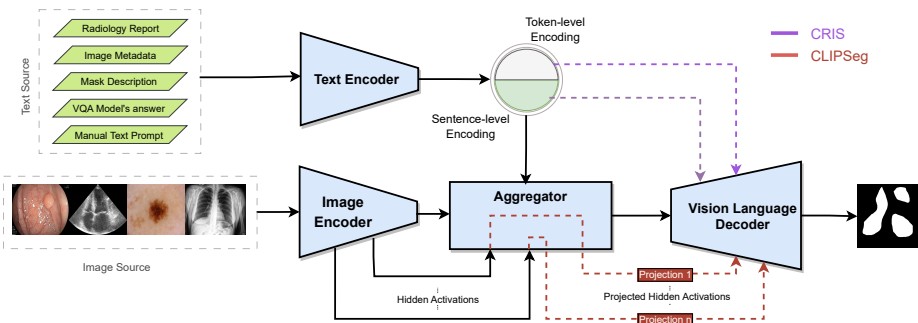

Figure 1: CRIS and CLIPSeg-variants include Text and Image encoders, an Aggregator, and a Vision-Language Decoder.

in both zero-shot and finetuning while only finetuning for BiomedCLIPSeg-based models as they lack an end-to-end pretrained encoder-decoder.

## 2.2. Datasets

We collected 11 2D medical imaging datasets of diverse modalities, organs, and pathologies covering both radiology and non-radiology images for binary and multi-class segmentation tasks (see Table 1). All the datasets are used for finetuning separately or combined (as a single pooled dataset) except the last three endoscopy datasets (ETIS, ColonDB, and CVC300), which are used only as the test split to study domain shift robustness.

Table 1: Datasets overview for single and multi-class segmentation tasks.

| Category | Modality | Organ | Name | Foreground Class(es) | # train/val/test |
|----------|----------|-------|------|----------------------|------------------|
| Non-Radiology | Endoscopy | Colon | Kvasir-SEG | Polyp | 800/100/100 |
| | | | ClinicDB | | 490/61/61 |
| | | | BKAI | | 800/100/100 |
| | | | ETIS | | 0/0/196 |
| | | | ColonDB | | 0/0/380 |
| | | | CVC300 | | 0/0/60 |
| | Photography | Skin | ISIC 2016 | Skin Lesion | 810/90/379 |
| | | Foot | DFU 2022 | Foot Ulcer | 1600/200/200 |
| Radiology | Ultrasound | Heart | CAMUS | Myocardium, Left ventricular, and Left atrium cavity | 4800/600/600 |
| | | Breast | BUSI | Benign and Malignant Tumors | 624/78/78 |
| | X-Ray | Chest | CheXlocalize | Atelectasis, Cardiomegaly, Consolidation, Edema, Enlarged Cardiomediastinum, Lung Lesion, Lung Opacity, Pleural Effusion, Pneumothorax, and Support Devices | 1279/446/452 |

## 2.3. Generating Language Prompts

Although language prompts enable injecting rich information into VLSMs, manually crafting individual image-specific prompts becomes impractical for large-scale evaluations. Thus, we implement an automated prompt generation system for extensive assessments of medical VLSMs. This involves incorporating semantic concepts such as size, position, color, and specific medical attributes like gender, age, and pathology.

In addition to automated prompts, we introduce manual prompts that provide general class-level information applicable to all samples within a given dataset. The generated language prompts encapsulate a comprehensive set of attributes and information, comprising: (**i**) Inspired by Tomar et al. (2022), *number*, *size*, and *relative location* are derived through image processing on segmentation masks. (**ii**) Motivated by Qin et al. (2022), we use *shape* and *color* information from VQA queries. (**iii**) *General class information*, extracted for photographic images from online medical journals, provides overarching details applicable across different datasets. Notably, Qin et al. (2022) used PubMedBERT (Gu et al., 2021) for this purpose; however, our experiments revealed its unreliability, leading us to manually gather this information from online medical journals (see Table 5). (**iv**) Attributes like *age*, *gender of patients*, *image quality*, *cardiac cycle*, and *tumor type* are extracted whenever available, contributing valuable context to the language prompts. There are 14 such attributes, (**a1** to **a14**), which we combined in various ways to build nine distinct prompt types (**P1** to **P9**) for each dataset (Table 9; Appendix G). Each prompt type caters to specific attribute combinations, prioritizing the class name as the foundational attribute and enhancing the versatility of the generated prompts.

### 2.4. Implementation Details

We finetuned VLSMs with minimal hyperparameter changes from the original pretraining settings. AdamW (Loshchilov and Hutter, 2017) optimizer with weight decay of $10^{-3}$, and initial learning rates of $2 \times 10^{-3}$ (CLIPSeg) and $2 \times 10^{-5}$ (CRIS) were utilized. Dice loss was used alongside Binary Cross Entropy loss scaled by 0.2. The learning rate was reduced by 10 times if validation loss did not decrease for 5 consecutive epochs. Batch sizes of 128 and 32 were used for CLIPSeg and CRIS, respectively, due to the difference in model sizes[2].

## 3. Results

**VLSMs adapt better to non-radiology images in Zero-Shot Setting (ZSS).** Both CRIS and CLIPSeg barely work in ZSS for radiology images except for CRIS in the BUSI dataset but get a Dice score in the range of $20\% - 70\%$ for non-radiology datasets, with $67.98\%$ being the highest Dice score for ISIC (Figure 2). Adding more attributes to the prompt generally improved performance, but the gain is inconsistent across prompts and datasets.

**Image-specific-attributes or general descriptions?** In the ZSS, CRIS performs better on endoscopy datasets when prompts contain image-specific attributes (*size*, *number*, and *location*; **P4**, **P5**, and **P6**; Figure 2), but degrades with non-image-specific attributes added (**P7**, **P8**, **P9**). Interestingly, prompts with general descriptions (**P8** and **P9**) achieve the highest performance on the DFU 2022 dataset, possibly due to pretrained models' familiarity with feet and skin compared to the colon. This highlights the complex relationship between pretraining data, VLSM architecture, and the medical segmentation task.

**Making prompts richer does not always help during finetuning.** Figure 2 shows that the DSC variation across prompt type is minimal in the finetuned setting for all

---

2. Further details are in Appendix C.

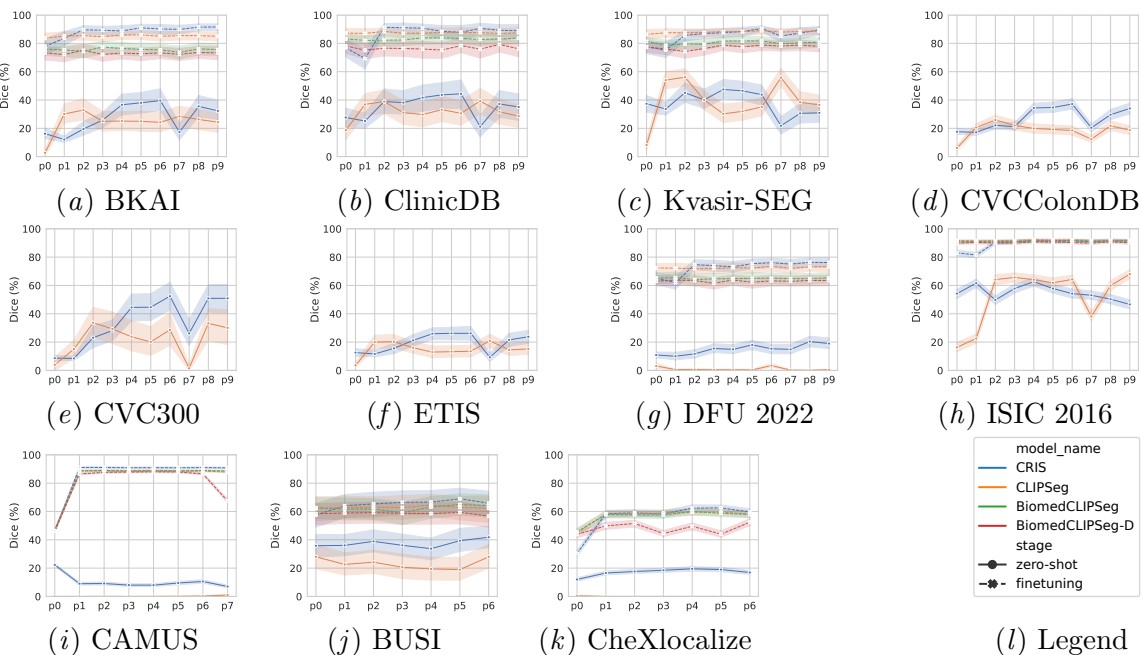

Figure 2: Zero-shot and finetuning performance of CRIS, CLIPSeg, BiomedCLIPSeg, and BiomedCLIPSeg-D model on non-radiology (first two rows) and radiology datasets (last row). Finetuning using the prompts improves performance compared to the empty prompt, particularly in multi-class settings.

the models. Prompt with only *class name* (**P1**) improves segmentation performance in radiology datasets for all four VLSMs. While CRIS' performance almost saturates after adding the *class name* and *mask shape* (**P2**), the rest of the models have similar performance for all the prompts except **P0** with multi-class segmentation (CAMUS and CheXlocalize).

BiomedCLIPSeg and BiomedCLIPSeg-D, despite being based on a VLM pretrained on medical data, consistently perform poorly across all prompts compared to CLIP and CLIPSeg. This is likely because it has not been further pretrained for segmentation tasks on a large-scale dataset. Subsequent experiments use better performing CLIPSeg and CRIS to study the impact of individual attributes and robustness of VLSMs[3].

**When finetuned, CRIS captures some language semantics better than CLIPSeg.** We replaced attribute values of the input prompts during inference with random uncommon English words and semantically wrong or opposite values to assess whether VLSMs leverage the language semantics. Figure 3 shows that altering attributes minimally impacts CLIPSeg's performance but notably deteriorates CRIS's. To further investigate CLIPSeg's indifference to attribute values, we provided only the *class name*(**P1**) as input during in-

---

3. Additionally, we have also trained both the models, keeping their encoders frozen whose results are shown in Appendix F.1.

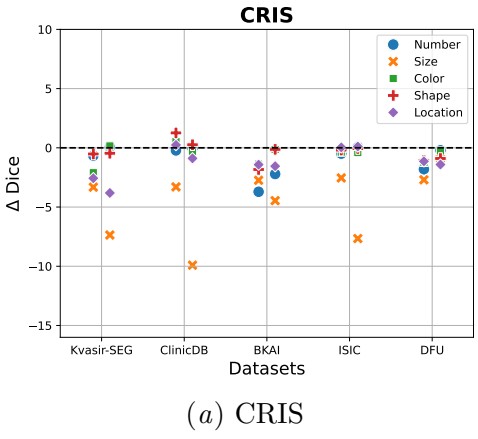 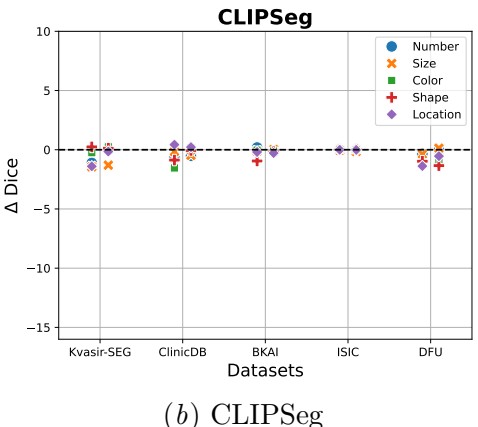

($a$) CRIS        ($b$) CLIPSeg

Figure 3: Relative change in percentage dice score on replacing attribute values by a random uncommon English word (left of vertical lines) or semantically opposite value such as replacing 'large' with 'small' (right of vertical lines) in prompt *P6*.

ference to the model trained on rich prompts **P6**; the results were very similar to providing the rich prompts, reinforcing the minimal impact of attributes in CLIPSeg.

CRIS's performance decreases notably for attributes like size and location. The decline is more significant when providing semantically opposite values than random uncommon English words, indicating robust semantic learning. A qualitative examination of predicted segmentation masks confirms this trend (Figure 4).

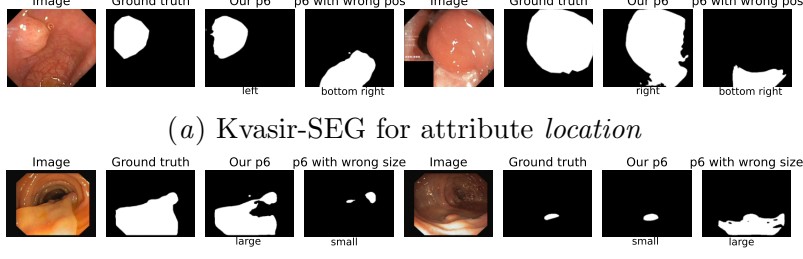

($a$) Kvasir-SEG for attribute *location*

($b$) ClinicDB for attribute *size*

Figure 4: Examples of images with the highest drops in dice score for two datasets when values for sensitive attributes are replaced with another value within the value set of the attributes in the dataset in *P6*.

**Finetuned VLSMs comparable to SOTA segmentation models.** Table 2 compares VLSMs vs. traditional CNN-based models (Ronneberger et al., 2015; Chen et al., 2018; Zhou et al., 2018) on their ability to learn in two scenarios: when trained on (**i**) individual specialized datasets or (**ii**) a pooled dataset that combines diverse datasets into a single

Table 2: Performance of VLSMs (Dice (%)) and CNN models when finetuning in different combinations of datasets. For each column, **Bold** and **Bold with underline** represent the best result among all models for the specific dataset combination and all combinations, respectively.

| Finetuned Dataset | Model | Kvasir-SEG | ClinicDB | BKAI | CVC-300 | CVC-ColonDB | ETIS | ISIC | DFU | CAMUS | BUSI | CheXlocalize |
|---|---|---|---|---|---|---|---|---|---|---|---|---|
| Individual | CRIS | **91.39** | **91.69** | **92.40** | - | - | - | 91.94 | **76.13** | **91.09** | 69.31 | **62.57** |
| | CLIPSeg | 89.51 | 88.74 | 86.47 | - | - | - | **92.12** | 73.24 | 88.85 | 64.32 | 59.56 |
| | UNet | 84.77 | 85.65 | 83.79 | - | - | - | 90.40 | 67.87 | 90.19 | **75.21** | 50.29 |
| | UNet++ | 84.70 | 84.16 | 84.61 | - | - | - | 90.12 | 69.95 | 89.95 | 72.55 | 49.53 |
| | DeepLabv3+ | 84.11 | 89.11 | 84.95 | - | - | - | 90.66 | 67.89 | 90.43 | 70.57 | 49.95 |
| | *SOTA** | 95.02 | 95.73 | 90.23 | - | - | - | 92.00 | 72.87 | 94.10 | 89.80 | - |
| Pooled | CRIS | 90.23 | 91.88 | 90.21 | 88.99 | 78.07 | 75.93 | 91.99 | 75.55 | 91.00 | 67.89 | 61.01 |
| | CLIPSeg | 87.25 | 87.49 | 87.30 | 87.24 | 71.32 | 69.64 | 91.34 | 71.94 | 88.76 | 66.02 | 56.60 |
| | UNet | 36.60 | 26.10 | 37.70 | 4.94 | 8.55 | 12.00 | 64.90 | 38.60 | 76.82 | 44.60 | 38.00 |
| | UNet++ | 80.52 | 78.21 | 77.87 | 87.80 | 51.92 | 48.16 | 88.41 | 65.78 | 89.99 | 75.59 | 53.88 |
| | DeepLabv3+ | 82.40 | 82.70 | 77.60 | 84.40 | 59.30 | 48.30 | 89.60 | 67.70 | 90.17 | **77.80** | 54.56 |
| Endoscopy Pooled | CRIS | **91.25** | **92.94** | **92.35** | **90.42** | **81.00** | **79.67** | - | - | - | - | - |
| | CLIPSeg | 89.62 | 88.96 | 86.98 | 88.98 | 75.23 | 71.18 | - | - | - | - | - |
| | UNet | 85.45 | 88.17 | 84.70 | 90.27 | 67.87 | 61.84 | - | - | - | - | - |
| | UNet++ | 83.99 | 85.44 | 82.27 | 89.4 | 66.61 | 55.62 | - | - | - | - | - |
| | DeepLabv3+ | 87.87 | 87.60 | 84.38 | 87.54 | 69.95 | 65.24 | - | - | - | - | - |
| | *SOTA Sources | Dumitru et al. (2023) | Fitzgerald and Matuszewski (2023) | Tomar et al. (2022) | - | - | - | Hasan et al. (2022) | Liao et al. (2022) | Ling et al. (2022) | Zhang et al. (2023b) | - |

training set. While the segmentation models (CNNs and VLSMs) achieve better on pooled endoscopy datasets than individual endoscopy datasets, performance mainly drops when training on a pooled set comprising all the datasets. VLSMs outperform image-only off-the-shelf CNN-based methods in most cases. We have also compared with the best method reported in the literature for each dataset.[4] The state-of-the-art results[5] are better, although VLSMs seem to have competitive performance.

**VLSMs adapt better to distribution shifts.** To assess the ability of the segmentation models to transfer knowledge learned from one dataset to another similar one, we train the models on each large endoscopy dataset (Kvasir-SEG, ClinicDB, and BKAI) and evaluate them on all endoscopy datasets. Table 3, shows that VLSMs perform better in all the cases than the conventional models for endoscopic datasets. VLSMs show smaller performance drops than conventional models when trained on a different distribution from the test set.

## 4. Discussion, Limitations, and Conclusion

VLSMs pretrained on natural images show suboptimal zero-shot accuracy with medical images for practical use but provide a foundation for joint text-image representation. Our study provides intriguing insights into prompt design, attributes' roles, and models' performance when finetuning across diverse datasets. The zero-shot segmentation performance showed improvement across all non-radiology datasets when compared to the radiology datasets. This could be attributed to the non-radiology medical imaging modalities being closer to open-domain images, as well as the potential familiarity with organs such as skin and feet (for ISIC and DFU datasets) during pretraining. The best-performing prompts

---

4. To ensure a thorough comparison across datasets with diverse modalities and SOTA methods, we report the SOTA for each dataset from literature, apart from implementing a few commonly used CNN baselines.
5. Except for CAMUS and ISIC, may have different training, validation, and test splits due to the unavailability of the standard splits in literature.

Table 3: Segmentation performance (Dice (%)) on out-of-distribution endoscopy datasets. For each column, **Bold** and **Bold with underline** show the best result across the model concerning the tested dataset for each finetuning dataset and across the finetuning datasets, respectively. The shaded results correspond to results in test sets of the same distribution, while the rest are on out-of-distribution test sets.

| Tested on → Finetuned on ↓ | Model ↓ | Kvasir-SEG | ClinicDB | BKAI | CVC-300 | CVC-ColonDB | ETIS |
|---|---|---|---|---|---|---|---|
| **Kvasir-SEG** | CRIS | **91.39** | **82.99** | **83.26** | 86.15 | **76.87** | **62.99** |
| | CLIPSeg | 89.51 | 80.21 | 77.89 | **86.49** | 70.46 | 62.83 |
| | UNet | 84.77 | 64.84 | 66.22 | 77.16 | 50.81 | 34.98 |
| | UNet++ | 84.70 | 68.15 | 61.76 | 79.35 | 52.3 | 32.81 |
| | DeepLabv3+ | 84.11 | 68.0 | 63.57 | 76.93 | 58.41 | 33.81 |
| **ClinicDB** | CRIS | 82.66 | **91.69** | 76.21 | **87.47** | 76.14 | 64.62 |
| | CLIPSeg | **84.02** | 88.74 | 72.04 | 87.07 | 67.91 | 60.09 |
| | UNet | 65.80 | 85.65 | 35.26 | 73.91 | 55.01 | 29.66 |
| | UNet++ | 61.93 | 84.16 | 38.81 | 71.15 | 55.05 | 23.16 |
| | DeepLabv3+ | 66.63 | 89.11 | 40.89 | 82.05 | 61.79 | 39.53 |
| **BKAI** | CRIS | **83.74** | **78.18** | **92.40** | 79.48 | **65.30** | 66.72 |
| | CLIPSeg | 83.70 | 76.07 | 86.47 | **86.06** | 63.59 | **66.97** |
| | UNet | 68.42 | 62.20 | 83.79 | 60.13 | 44.52 | 42.91 |
| | UNet++ | 70.64 | 62.66 | 84.61 | 82.44 | 55.60 | 46.84 |
| | DeepLabv3+ | 69.02 | 61.99 | 84.95 | 77.47 | 53.15 | 49.61 |

vary with datasets but often include attributes familiar to models during pretraining. For instance, CRIS trained on RefCOCO (Kazemzadeh et al., 2014) for referring image segmentation captures size, location, and number well.

The ability of CRIS to leverage better language semantics than CLIPSeg might be due to (**i**) CRIS's architecture that focuses on token-level intervention instead of CLIPSeg's sentence-level embedding, and (**ii**) end-to-end VLSM training of CRIS compared to CLIPSeg's training for segmentation task with frozen CLIP encoder. Interestingly, models based on CLIP performing better than those based on BiomedCLIP (pretrained with image-text pairs of 400 million natural domain versus 15 million medical domain) shows that large-scale dataset has the benefit that is hard to achieve with smaller-scale domain-specific data.

Our study aims to build insights into how well VLSMs leverage textual information and perform transfer learning in the medical domain. It proposes pragmatic prompt settings and systematic experiments instead of implementing an exhaustive list of VLSMs and only grossly comparing their performance. The four CLIP-based VLSMs cover significant variations in architecture to capture global vs. token level information in prompts, training approach with end-to-end for referring image segmentation vs. finetuning only decoder for segmentation, and based on VLM pretrained on natural vs. medical domain, etc. We focus only on 2D medical images, excluding 3D modalities like MRI or CT scans, as most existing VLSMs are suitable only for 2D images, requiring further research in building 3D VLSMs.

While the VLSMs' performance seems on par with image-only architectures, and some of the VLSMs use information injected via text prompts, our results show that further research is needed to develop novel approaches that can better leverage the rich information provided via prompts. Moreover, interesting future directions can explore how these prompts could help build more robust and explainable models against out-of-distribution data. Our work serves as an essential first step in this direction, offering a valuable evaluation framework, datasets enriched with prompts, and fascinating insights for future investigation.

## Acknowledgments

We thank Kathmandu University for their invaluable support in granting us access to their supercomputer infrastructure. This enabled the successful execution of the experiments crucial to this paper.

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

## Appendix A. Impact on Society

The incorporation of language prompts in medical image segmentation has the potential to impact society significantly, particularly in clinical settings. By enabling radiologists to quickly and accurately segment complex shapes using just a few words, language prompts offer a more interpretable and explainable approach compared to traditional visual prompts such as points or boxes.

One significant advantage of language prompts is their ability to convey detailed information about normal and abnormal structures' texture, shape, and spatial relationships. This allows for a more comprehensive understanding of medical images, facilitating more accurate segmentation results. Additionally, language prompts can be easily adapted to new classes, making them highly versatile and adaptable in various medical scenarios.

Using language prompts in medical image segmentation can improve the efficiency and effectiveness of radiologists' work, potentially leading to faster diagnoses and treatment decisions. Moreover, the interpretability of language prompts can aid in building trust and confidence among healthcare professionals and patients as the reasoning behind the segmentation process becomes more transparent.

Overall, the integration of language prompts in medical image segmentation has the potential to revolutionize clinical practices, providing radiologists with a powerful tool to enhance their segmentation capabilities and ultimately improve patient care outcomes.

We strongly encourage and invite other researchers to contribute to this field of study. This research paper has no negative impact on society or further research in medical imaging, as we have adhered to ethical considerations in medical imaging and have not expressed disapproval of any previous studies.

## Appendix B. Dataset and Code Access

The GitHub repository[6] contains the source code with detailed documentation, the generated prompts for all the datasets, and thorough instructions along with the relevant links to access the individual image-mask pair datasets used in this work.

## Appendix C. Experiments

### C.1. VLSM Finetuning Experiments

CLIPSeg and CRIS internally resize the three-channeled input images to $352 \times 352$ and $416 \times 416$, respectively. The dice scores mentioned in the paper are calculated after resizing the output of the models back to the original size (before respective resizing). We normalize the resized images with means and standard deviations provided by the respective models and haven't performed other preprocessing and post-processing to access the models' raw performance.

For the five non-radiology datasets (Kvasir-SEG, ClinicDB, BKAI, ISIC, and DFU), we finetune VLSMs with ten prompts for an individual dataset, resulting in 50 experiments for each VLSM. Similarly, in the case of radiology datasets (CAMUS, BUSI, and CheXlocalize), we have a total of 22 finetuning experiments for each VLSM. We also finetune CRIS and

---

6. https://github.com/naamiinepal/medvlsm

CLIPSeg with the pooled datasets comprising only endoscopic and all datasets. Thus, including all varieties with the VLSMs and the different prompting mechanisms, we have 442 finetuning experiments.

The average time to fine-tune CRIS for a dataset on a prompt is approximately 60 minutes in our training setup, running 45 epochs on average. For CLIPSeg, the average training time is 40 minutes, running for 90 epochs on average. BiomedCLIPSeg's and BiomedCLIPSeg-D's average training times are 20 minutes and 30 minutes, running for 80 epochs and 50 epochs, respectively. We monitored the segmentation metric on the held-out validation sets for early stopping, with patience of 50 epochs for CLIPSeg variants and 10 epochs for CRIS.

### C.2. Hyperparameter Tuning

We experiment with multiple sets of hyperparameters including learning rates, optimizers, batch sizes, and schedulers. We select the optimal setting of hyperparameters (as mentioned in the main paper) that showed optimal performance in most datasets (Table 4).

| | |
|---:|:---|
| Optimizers | {Adam, AdamW} |
| Learning Rates (LRs) | $[10^{-5}, 10^{-2}]$ |
| LR Schedulers | {CosineAnealingLR, ConstantLR, ReduceLROnPlateau} |
| Batch sizes | $\{16, 32, 64, 128\}$ |

Table 4: Different settings of hyperparameters that have been experimented with to select the optimal one.

### C.3. CNN-based Experiments

For comparative analysis, we consider three of the conventional CNN-based segmentation models: UNet (Ronneberger et al., 2015), UNet++(Zhou et al., 2018), and DeepLabV3+ (Chen et al., 2018). For all of the models, we use pretrained ResNet-50 (He et al., 2016) as the backbone, and default parameters given by the framework *Segmentation Models PyTorch*[7] are chosen as the model hyperparameters. We use Dice loss for error propagation within the models with Adam optimizer (Kingma and Ba, 2014) of learning rate $10^{-3}$ and zero weight decay.

## Appendix D. PubMedBERT's failure to give reliable output

Table 5 contains the predictions of PubMedBERT for the masked language modeling in different datasets.

## Appendix E. Some visualizations and qualitative analysis

Some visualizations and qualitative analysis are shown in Figures 5 and 6.

---

7. https://github.com/qubvel/segmentation_models.pytorch

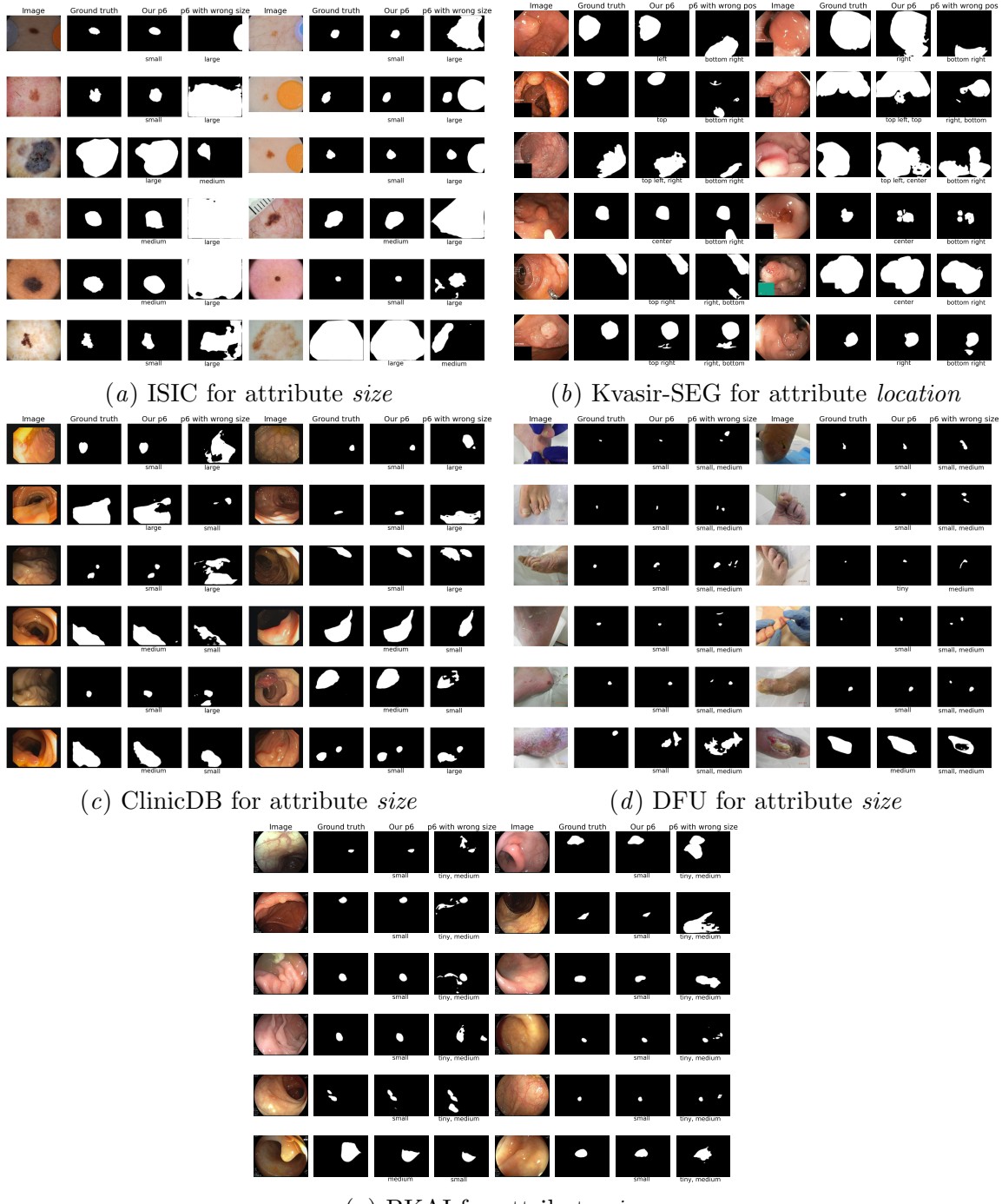

(a) ISIC for attribute *size*

(b) Kvasir-SEG for attribute *location*

(c) ClinicDB for attribute *size*

(d) DFU for attribute *size*

(e) BKAI for attribute *size*

Figure 5: Visualization of CRIS's performance when prompt attributes are changed using a wrong attribute value. For each medical image, three corresponding masks are displayed: ground truth mask, output mask for the corresponding prompt, and output mask after altering an attribute value of the prompts.

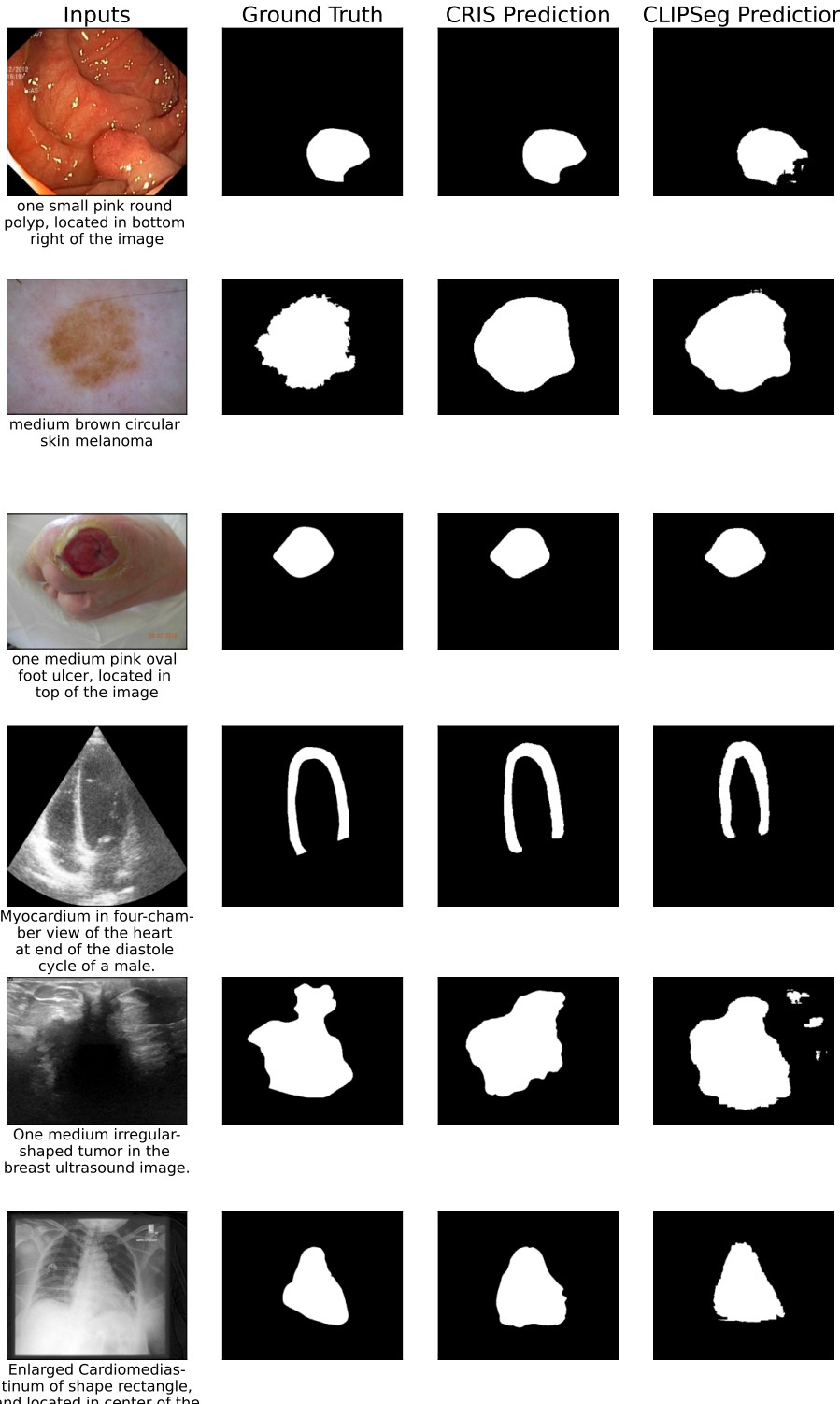

Figure 6: Sample input, ground truth, and models' predictions

Table 5: PubMedBERT's top five predictions for the masked language modeling inference. The predictions are ordered in the descending order of the probability generated by the model. The model has high uncertainty as the maximum probability is about 0.1. The predictions are almost the same and uninformative, which is more prominent in the radiology datasets.

| Dataset | Masked sentence | Top-5 Predictions |
|---|---|---|
| All Endoscopy* | The location of the polyp is [MASK]. | [variable, unknown, varied, unpredictable, uncertain] |
| | Polyp is located at [MASK]. | [bifurcation, apex, rectum, midline, right] |
| | The shape of polyp is [MASK]. | [irregular, variable, oval, round, different] |
| | Polyp is [MASK] in shape. | [oval, irregular, round, spherical, cylindrical] |
| | The color of the polyp is [MASK]. | [yellow, red, blue, brown, pink] |
| | Polyp is [MASK] in color. | [yellow, white, red, black, green] |
| ISIC | The location of skin melanoma is [MASK]. | [unknown, variable, unusual, unpredictable, rare] |
| | The color of skin melanoma is [MASK]. | [red, yellow, brown, black, blue] |
| | Skin melanoma is [MASK] in texture. | [heterogeneous, variable, soft, irregular, fibrous] |
| | Skin cancer is located at [MASK]. | [extremities, birth, puberty, adolescence, skin] |
| | Skin cancer is [MASK] in texture. | [heterogeneous, unique, variable, diverse, distinctive] |
| DFU | The location of a diabetic foot ulcer is at [MASK]. | [first, rest, ankle, home, foot] |
| | Diabetic foot ulcer is located at [MASK]. | [ankle, heel, foot, extremities, feet] |
| | The location of the foot ulcer is [MASK]. | [ankle, knee, first, heel, night] |
| | Foot ulcer is located at [MASK]. | [ankle, heel, foot, knee, night] |
| CAMUS | The left ventricular cavity is [MASK] in shape. | [spherical, triangular, normal, oval, round] |
| | The myocardium is [MASK] in shape. | [spherical, cylindrical, circular, round, triangular] |
| | The left atrium cavity is [MASK] in shape. | [oval, round, triangular, spherical, irregular] |
| | The left ventricular cavity is located at [MASK]. | [diastole, apex, rest, 90°, 45°] |
| | The myocardium is located at [MASK]. | [rest, apex, risk, diastole, birth] |
| | The left atrium cavity is located at [MASK]. | [diastole, right, left, 90°, apex] |
| BUSI | The malignant breast tumor is [MASK] in shape. | [round, irregular, oval, solid, spherical] |
| | The benign breast tumor is [MASK] in shape. | [oval, round, irregular, solid, spherical] |
| CheXlocalize | Airspace Opacity is [MASK] in shape. | [irregular, oval, round, triangular, globular] |
| | Enlarged Cardiomediastinum is [MASK] in shape. | [oval, triangular, irregular, round, rounded] |
| | Cardiomegaly is [MASK] in shape. | [irregular, triangular, normal, oval, round] |
| | Lung Opacity is [MASK] in shape. | [irregular, round, oval, nodular, reticular] |
| | Consolidation is [MASK] in shape. | [spherical, circular, triangular, irregular, round] |
| | Atelectasis is [MASK] in shape. | [irregular, oval, triangular, spherical, round] |
| | Pleural Effusion is [MASK] in shape. | [irregular, round, oval, spherical, solid] |

*This includes six datasets of endoscopy: Kvasir-SEG, ClinicDB, BKAI, CVC-300, CVC-ColonDB, ETIS

# Appendix F. Results

## F.1. Finetuning only the Decoders for CLIP-based VLSMs

Tables 6 and 7 show the results of VLSMs with finetuned the decoder while keeping the encoders frozen.

## F.2. Using radiology reports for lung segmentation

To examine the usage of free-text radiology reports of chest x-rays for segmentation, we utilize 1,141 frontal-view CXRs randomly selected from the MIMIC-CXR database (Johnson et al., 2019a,b; Chen et al., 2022). This dataset contains the segmentation of lungs, which has been verified manually. We use the free-text radiology reports provided in the MIMIC-CXR Database (Johnson et al., 2019a) as the only prompt (P1), and the results are reported in Table 8.

Table 6: Finetuned segmentation Dice score (%) of CRIS on different datasets on different sets of prompts with frozen CLIP.

| Prompt → / Dataset ↓ | P0 | P1 | P2 | P3 | P4 | P5 | P6 | P7 | P8 | P9 |
|---|---|---|---|---|---|---|---|---|---|---|
| Kvasir-SEG | $75.49_{\pm27.22}$ | $76.03_{\pm26.35}$ | $82.18_{\pm22.40}$ | $81.89_{\pm21.78}$ | $84.26_{\pm20.39}$ | $\mathbf{86.39_{\pm17.01}}$ | $85.37_{\pm17.29}$ | $82.43_{\pm22.11}$ | $85.06_{\pm18.89}$ | $85.02_{\pm19.10}$ |
| ClinicDB | $49.48_{\pm33.67}$ | $46.98_{\pm34.30}$ | $81.07_{\pm24.37}$ | $82.72_{\pm23.96}$ | $84.88_{\pm24.00}$ | $85.01_{\pm21.84}$ | $83.31_{\pm22.74}$ | $81.66_{\pm26.10}$ | $\mathbf{87.13_{\pm21.38}}$ | $84.65_{\pm22.25}$ |
| BKAI | $77.98_{\pm28.73}$ | $75.01_{\pm29.97}$ | $81.93_{\pm24.66}$ | $82.49_{\pm24.7}$ | $82.39_{\pm23.65}$ | $84.65_{\pm21.75}$ | $85.75_{\pm21.48}$ | $84.91_{\pm23.06}$ | $\mathbf{86.40_{\pm20.49}}$ | $85.07_{\pm22.01}$ |
| ISIC | $87.64_{\pm14.37}$ | $85.77_{\pm18.29}$ | $90.25_{\pm10.37}$ | $90.32_{\pm10.93}$ | $91.28_{\pm7.45}$ | $91.23_{\pm8.56}$ | $\mathbf{91.29_{\pm8.10}}$ | $90.46_{\pm10.90}$ | $91.29_{\pm8.11}$ | $91.28_{\pm7.65}$ |
| DFU | $66.30_{\pm29.57}$ | $66.14_{\pm29.81}$ | $\mathbf{70.28_{\pm27.11}}$ | $67.24_{\pm30.22}$ | $69.19_{\pm28.98}$ | $68.55_{\pm29.56}$ | $68.93_{\pm29.41}$ | $69.35_{\pm28.75}$ | $68.36_{\pm29.62}$ | $70.15_{\pm28.59}$ |
| CAMUS | $46.15_{\pm9.69}$ | $88.87_{\pm8.49}$ | $\mathbf{89.18_{\pm6.79}}$ | $88.94_{\pm7.05}$ | $88.92_{\pm6.69}$ | $88.02_{\pm7.37}$ | $88.96_{\pm6.84}$ | $89.04_{\pm6.85}$ | N/A | N/A |
| BUSI | $47.11_{\pm39.12}$ | $61.49_{\pm36.03}$ | $63.18_{\pm36.89}$ | $62.87_{\pm37.60}$ | $65.10_{\pm36.60}$ | $66.69_{\pm35.68}$ | $\mathbf{66.76_{\pm35.77}}$ | N/A | N/A | N/A |
| CheXlocalize | $41.03_{\pm24.96}$ | $54.18_{\pm25.77}$ | $54.57_{\pm25.06}$ | $53.30_{\pm25.16}$ | $\mathbf{56.17_{\pm24.73}}$ | $56.03_{\pm24.49}$ | $52.48_{\pm25.89}$ | N/A | N/A | N/A |

Table 7: Finetuned segmentation Dice score (%) of CLIPSeg on different datasets on different sets of prompts with frozen CLIP.

| Prompt → / Dataset ↓ | P0 | P1 | P2 | P3 | P4 | P5 | P6 | P7 | P8 | P9 |
|---|---|---|---|---|---|---|---|---|---|---|
| Kvasir-SEG | $86.38_{\pm17.8}$ | $87.50_{\pm15.35}$ | $87.49_{\pm14.29}$ | $87.68_{\pm14.60}$ | $88.33_{\pm10.95}$ | $88.25_{\pm12.11}$ | $\mathbf{88.98_{\pm11.98}}$ | $87.97_{\pm13.93}$ | $88.39_{\pm14.72}$ | $88.71_{\pm11.4}$ |
| ClinicDB | $87.23_{\pm14.93}$ | $87.07_{\pm14.43}$ | $\mathbf{88.41_{\pm11.01}}$ | $87.17_{\pm14.73}$ | $87.25_{\pm15.09}$ | $87.73_{\pm13.52}$ | $87.76_{\pm13.56}$ | $87.57_{\pm13.98}$ | $87.05_{\pm14.79}$ | $87.46_{\pm14.39}$ |
| BKAI | $83.64_{\pm18.59}$ | $85.26_{\pm15.40}$ | $85.47_{\pm15.15}$ | $84.7_{\pm16.94}$ | $85.93_{\pm14.66}$ | $\mathbf{86.01_{\pm14.84}}$ | $85.02_{\pm17.23}$ | $85.45_{\pm14.76}$ | $85.50_{\pm15.68}$ | $84.99_{\pm17.11}$ |
| ISIC | $91.71_{\pm8.68}$ | $91.45_{\pm8.47}$ | $91.66_{\pm8.29}$ | $91.85_{\pm8.36}$ | $\mathbf{92.11_{\pm6.87}}$ | $92.02_{\pm6.88}$ | $92.09_{\pm7.00}$ | $91.77_{\pm7.73}$ | $91.89_{\pm7.70}$ | $91.90_{\pm7.21}$ |
| DFU | $72.35_{\pm25.04}$ | $72.19_{\pm25.69}$ | $71.79_{\pm25.05}$ | $71.88_{\pm24.83}$ | $72.5_{\pm24.43}$ | $72.31_{\pm25.27}$ | $\mathbf{73.53_{\pm23.68}}$ | $72.1_{\pm25.48}$ | $73.11_{\pm23.98}$ | $73.31_{\pm23.81}$ |
| CAMUS | $46.48_{\pm9.07}$ | $88.67_{\pm6.25}$ | $88.70_{\pm5.93}$ | $\mathbf{88.81_{\pm6.15}}$ | $88.77_{\pm6.22}$ | $88.47_{\pm6.55}$ | $88.53_{\pm6.29}$ | $87.82_{\pm7.01}$ | N/A | N/A |
| BUSI | $62.03_{\pm38.3}$ | $62.79_{\pm37.55}$ | $62.97_{\pm37.27}$ | $62.85_{\pm36.66}$ | $\mathbf{64.47_{\pm37.54}}$ | $62.83_{\pm38.19}$ | $62.33_{\pm38.68}$ | N/A | N/A | N/A |
| CheXlocalize | $45.35_{\pm25.18}$ | $58.10_{\pm25.03}$ | $58.37_{\pm24.50}$ | $58.95_{\pm24.48}$ | $59.49_{\pm25.11}$ | $\mathbf{59.56_{\pm24.70}}$ | $58.06_{\pm25.34}$ | N/A | N/A | N/A |

Table 8: Zero-shot and finetuning Dice scores (%) of the CRIS and CLIPSeg Manually labeled Chest X-ray Segmentation Dataset. We have used the actual radiology reports as **P1**. P0 indicates an empty prompt.

| Models ↓ | Experiment ↓ / Prompt → | P0 | P1 |
|---|---|---|---|
| CRIS | Zero-shot | $44.8_{\pm18.97}$ | $40.73_{\pm18.95}$ |
| | Finetuning | $81.66_{\pm5.65}$ | $90.99_{\pm1.41}$ |
| CLIPSeg | Zero-shot | $0.26_{\pm2.35}$ | $0.09_{\pm0.88}$ |
| | Finetuning | $91.39_{\pm1.09}$ | $91.22_{\pm1.26}$ |

## Appendix G.  Prompt Composition

The prompts used during the training for various datasets are shown below. If there are multiple templates for the same prompts for a dataset, one is randomly chosen during the training to increase the regularization for the models.

Table 9: Different prompts are formed for each dataset using combinations of 14 potential attributes. Although some attributes, like *Pathology*, are specific to some particular datasets, others, like *Class Keywords*, are common to all the datasets.

Attributes → **a1:** Class Keyword; **a2:** Shape; **a3:** Color; **a4:** Size; **a5:** Number; **a6:** Location; **a7:** General Class Info; **a8:** View; **a9:** Pathology; **10:** Cardiac Cycle; **a11:** Gender; **a12:** Age; **a13:** Image Quality; **a14:** Tumor Type

| Prompts → / Datasets ↓ | P1 | P2 | P3 | P4 | P5 | P6 | P7 | P8 | P9 |
|---|---|---|---|---|---|---|---|---|---|
| **Non-Radiology** | a1 | a1a2 | a1a2a3 | a1a2a3a4 | a1a2a3a4a5 | a1a2a3a4a6 | a1a7 | a1a2a3a4a5a7 | a1a2a3a4a5a6a7 |
| Example Prompt | **P9** → **one small pink round polyp** which is **often a bumpy flesh in rectum** located in **center** of the image | | | | | | | | |
| **CheXlocalize** | a1 | a1a8 | a1a2a8 | a1a2a6a8 | a1a2a6a8a9 | a1a9 | N/A | N/A | N/A |
| Example Prompt | **P5** → **Airspace Opacity** of shape **rectangle**, and located in **right** of the **frontal** view of a Chest Xray. **Enlarged Cardiomediastinum, Cardiomegaly, Lung Opacity, Consolidation, Atelectasis, Pleural Effusion** are present. | | | | | | | | |
| **CAMUS** | a1 | a1a8 | a1a8a10 | a1a8a10a11 | a1a8a10a11a12 | a1a8a10a11a12a13 | a1a8a10a11a12a13a2 | N/A | N/A |
| Example Prompt | **P7** → **Left ventricular cavity** of **triangular shape** in **two-chamber view** in the cardiac ultrasound at the end of the **diastole cycle** of a **40-year-old female** with **poor image quality**. | | | | | | | | |
| **BUSI** | a1 | a1a14 | a1a14a5 | a1a14a5a4 | a1a14a5a4a6 | a1a14a5a4a6a2 | N/A | N/A | N/A |
| Example Prompt | **P6** → **Two medium square-shaped benign tumors** at the **center, left** in the breast ultrasound image. | | | | | | | | |

## G.1. Non-radiology images

### G.1.1. Endoscopy Datasets

A total of six endoscopy datasets (polyp segmentation image-mask pairs) have been used for finetuning and evaluating our proposed models: Kvasir-SEG (Jha et al., 2020), ClinicDB (Bernal et al., 2015), BKAI (Ngoc Lan et al., 2021; An et al., 2022), CVC-300 (Vázquez et al., 2017), CVC-ColonDB (Tajbakhsh et al., 2015), and ETIS (Silva et al., 2014). The last three datasets have a small number of image-masks pairs, so they are used only for testing and evaluating the trained models.

1. **P0**: "" (No prompt)

2. **P1**: "*class name*"

   - *polyp*

3. **P2**: "*shape class name*"

   - *round polyp*

4. **P3**: "*color shape class name*"

   - *pink round polyp*

5. **P4**: "*size color shape class name*"

   - *medium pink round polyp*

6. **P5**: "*number size color shape class name*"

   - *one medium pink round polyp*

7. **P6**: "*number size color shape class name*, located in the *location* of the image"

   - *one medium pink round polyp*, located in the *top left* of the image

8. **P7**: "*class name*, which is a *general description of the class*"

   - *polyp*, which is a *small lump in the lining of colon*

9. **P8**: "*number size color shape class name*, which is a *general description of the class*"

   - *one medium pink round polyp*, which is a *small lump in the lining of colon*

10. **P9**: "*number size color shape class name*, which is a *general description of the class* located in the *location* of the image* "

    - *one medium pink round polyp*, which is a *small lump in the lining of colon* located in the *top left* of the image

For *General Description of the class*, prompts were built using information about the subject on the internet. Five such descriptions were designed for each dataset, and one random sample was selected each time as the *general description of the class* attribute whenever the prompts **p7**, **p8**, and **p9** were used.

G.1.2. ISIC and DFU-2022

The templates of prompts for the DFU-2022 (Kendrick et al., 2022) and ISIC (Gutman et al., 2016) datasets used were the same as the above examples for endoscopy images, with *class name* and *general description of the class* being different. We used class names **skin melanoma** and **foot ulcer** for the two datasets, respectively.

The five *General Description of the class* for each of the three types of photographic datasets used is listed in the table below.

Table 10: General Descriptions selected for each of the photographic datasets.

| Endoscopy Datasets | ISIC | DFU-2022 |
|---|---|---|
| → a projecting growth of tissue | → a spot with dark speckles | → a wound in foot and toes |
| → often a bumpy flesh in rectum | → a spot with irregular texture | → a sore in foot and toes |
| → a small lump in the lining of colon | → a dark sore with irregular texture | → a sore in skin of foot and toe |
| → a tissue growth that often resemble mushroom-like stalks | → an irregular sore with speckles | → an abnormality in foot and toes |
| → an abnormal growth of tissues projecting from a mucous membrane | → a rough wound on skin | → an open sore or lesion in foot and toes |

### G.2. Radiology Images

G.2.1. CHEXLOCALIZE

The prompts for the CheXlocalize (Saporta et al., 2022) dataset are listed below.

1. **P0**: "" (No prompt)

2. **P1**: "*labels* in a chest Xray."

   - *Airspace Opacity* in a chest Xray.

3. **P2**: "*labels* in the *xray_view* view of a Chest Xray."

   - Airspace Opacity in the *frontal* view of a Chest Xray.

4. **P3**: "*labels* of shape *shape* in the *xray_view* view of a Chest Xray."

   - Airspace Opacity of shape *rectangle* in the frontal view of a Chest Xray.

5. **P4**: "*labels* of shape *shape*, and located in *location* of the *xray_view* view of a Chest Xray."

   - Airspace Opacity of shape rectangle, and located in *right* of the frontal view of a Chest Xray.

6. **P5**: "*labels* of shape *shape*, and located in *location* of the *xray_view* view of a Chest Xray. *pathology* are present."

   - Airspace Opacity of shape rectangle, and located in right of the frontal view of a Chest Xray. *Enlarged Cardiomediastinum, Cardiomegaly, Lung Opacity, Consolidation, Atelectasis, Pleural Effusion* are present.

7. **P6**: "*labels* in a Chest Xray. *pathology* are present."

   - Airspace Opacity in a Chest Xray. Enlarged Cardiomediastinum, Cardiomegaly, Lung Opacity, Consolidation, Atelectasis, Pleural Effusion are present.

G.2.2. CAMUS

The prompts for the CAMUS (Leclerc et al., 2019) dataset are listed below.

1. Class of Current Image

   - *Left ventricular cavity*, *Myocardium*, or *Left atrium cavity* of the heart
   - [*class*] in the cardiac ultrasound

2. Include the chamber information

   - Left ventricular cavity in *two-chamber view* of the heart.
   - Left ventricular cavity in *two-chamber view* in the cardiac ultrasound.

3. Include the cycle

- Left ventricular cavity in two-chamber view of the heart at the *end of the diastole cycle.*
- Left ventricular cavity in two-chamber view in the cardiac ultrasound at the *end of the diastole cycle.*

4. Include the gender

- Left ventricular cavity in two-chamber view of the heart at the end of the diastole cycle of *a female.*
- Left ventricular cavity in two-chamber view in the cardiac ultrasound at the end of the diastole cycle of *a female.*

5. Include the age

- Left ventricular cavity in two-chamber view of the heart at the end of the diastole cycle of a *forty-six-year-old* female.
- Left ventricular cavity in two-chamber view in the cardiac ultrasound at the end of the diastole cycle of a *forty-six-year-old* female.

6. Include the image quality

- Left ventricular cavity in two-chamber view of the heart at the end of the diastole cycle of a 40-year-old female with *poor image quality.*
- Left ventricular cavity in two-chamber view in the cardiac ultrasound at the end of the diastole cycle of a 40-year-old female with *poor image quality.*

7. Include the mask shape

- Left ventricular cavity of *triangular shape* in two-chamber view of the heart at the end of the diastole cycle of a 40-year-old female with *poor image quality.*
- Left ventricular cavity of *triangular shape* in two-chamber view in the cardiac ultrasound at the end of the diastole cycle of a 40-year-old female with *poor image quality.*

G.2.3. Breast Ultrasound Images Dataset

The prompts for the Breast Ultrasound Images (BUSI) (Al-Dhabyani et al., 2020) dataset are listed below.

1. Presence of tumor

- *[No] tumor* in the breast ultrasound image

2. Tumor Type

- *Benign* tumor in the breast ultrasound image
- *Regular-shaped* tumor in the breast ultrasound image

3. Tumor Number

- *Two* benign tumors in the breast ultrasound image
- *Two* regular-shaped tumors in the breast ultrasound image

4. Tumor Coverage

- Two *medium* benign tumors in the breast ultrasound image
- Two *medium* regular-shaped tumors in the breast ultrasound image

5. Tumor Location

- Two medium benign tumors *at the center, left* in the breast ultrasound image
- Two medium regular-shaped tumors *at the center, left* in the breast ultrasound image

6. Tumor Shape

- Two medium *square-shaped* benign tumors at the center, left in the breast ultrasound image
- Two medium *square-shaped* regular tumors at the center, left in the breast ultrasound image

