# OpenReview forum: "Exploring Transfer Learning in Medical Image Segmentation using Vision-Language Models"
_MIDL.io/2024/Conference — MIDL 2024 Oral_

### Official Review · Reviewer_t1ep · 2024-02-29

**Confidence:** 3
**Preliminary Rating:** 2
**Recommendation:** Poster
**Final Rating:** 5

**Summary:**

This work is a study on transferring Vision-Language Segmentation Models (VLSMs) to 2D medical images, leveraging advancements in foundation Vision-Language Models (VLMs) used in natural image-text pair analysis. The study evaluates VLSMs, adapted from the popular CLIP and BiomedCLIP models, across 11 carefully selected medical image datasets spanning diverse modalities. The study also highlights VLSMs' enhanced performance and potential robustness against domain shifts when handling pooled datasets from different modalities. The paper details the implementation of automated and manual prompt generation, the finetuning process, and an extensive evaluation of VLSMs' robustness, performance, and adaptability to domain shifts. It concludes with a discussion on the limitations and future directions for leveraging VLSMs in medical image segmentation, emphasizing the need for further research to optimize the integration of textual information for improved segmentation outcomes.

**Strengths:**

The paper represents an important study on transferring VLSMs to 2D medical image segmentation.

The study encompasses a broad range of 11 curated medical image datasets covering diverse modalities, organs, and pathologies. This diversity is encouraging for the reader to evaluate performance  across different aspects of medical imaging.

The paper conducts thorough experiments with four different VLSMs, providing a deep dive into how these models perform in various scenarios, including zero-shot settings, fine-tuning, and their ability to handle domain shifts and pooled datasets. The models' capabilities and limitations are clearly established.

It is encouraging to see that the work is being open-sourced promoting transparency and reproducibility.

**Weaknesses:**

The paper does not compare VLSMs against the latest state-of-the-art image-only segmentation models or other emerging multimodal approaches, potentially limiting the context for evaluating VLSMs' performance.

The study leaves open questions about the optimal ways to integrate textual information into segmentation models, suggesting that further research is needed to develop methodologies that can fully exploit this aspect.

The study exclusively focuses on 2D medical images, excluding 3D modalities such as MRI or CT scans. This limitation restricts the applicability of the findings across the full spectrum of medical imaging technologies. The authors have listed the same as a limitation of the work.

**Detailed Comments:**

No additional comments.

**Justification Of Final Rating:**

I am increasing my initial rating to a strong accept. The authors have provided convincing arguments that helped me make this decision. Several of my concerns have been satisfactorily addressed by the authors.

**Justification Of The Preliminary Rating:**

The exploration of transferring VLSM's for medical imaging is of vital importance and the study provides some insight into the performance of VLSM's with transfer learning. However, the work is in preliminary stages and does not provide a well-rounded evaluation of the VLSM's. For instance, the study encompasses 11 distinct datasets, but does not discuss the unique challenges that different modalities pose for performance of the models. The motivation for segmentation in medical imaging is discussed in the paper, however, the discussion on the benefits of using VLSM over image-only segmentation models has not been included in the paper.

**Questions To Address In The Rebuttal:**

Insights into modality-specific challenges and opportunities could guide future research towards modality-adaptive approaches. Discussion on how different imaging modalities affect the performance of VLSMs would be a valuable information for the reader.

I think it is important to briefly discuss any ethical considerations related to the use of VLSMs in medical image analysis, including privacy concerns with patient data.

**Special Issue:**

No

---

> ### Author Response · Authors · 2024-03-18
> **Official Comment by Authors (1/2)**
>
> Thank you for your review and valuable feedback on our work.
>
> > **The paper does not compare VLSMs against the latest state-
> of-the-art image-only segmentation models or other emerging
> multimodal approaches, potentially limiting the context for
> evaluating VLSMs’ performance.**
>
> We compare the VLSMs against the latest state-of-the-art image-only segmentation models as baselines. Given the diverse range of data modalities and
> variations in the state-of-the-art methods across different datasets, we have reported the best method reported in the literature for each dataset in Table 2,
> row 6 to ensure comprehensive comparisons. We will add a line in the main
> manuscript to simplify this. Additionally, we implement a
> few of the most commonly used image-only medical segmentation models as baselines for further comparisons: UNet [7], UNet++ [8], and DeepLabv3+ [9]
> in Tables 2 and 3.
>
> > **The study leaves open questions about the optimal ways to
> integrate textual information into segmentation models, suggesting that further research is needed to develop methodologies that can fully exploit this aspect.**
>
> We agree with the reviewer that the study leaves open questions on optimal
> ways to integrate textual information into segmentation models, but we beg to
> disagree that this is the weakness of this study; instead, we think this is the
> strength of this work, which through rigorous and comprehensive experimental
> designs showed this gap in the literature and SOTA VLSMs, providing the
> community one important direction to explore and evaluate new VLSMs that
> gets proposed.
>
> > **The study exclusively focuses on 2D medical images, excluding 3D modalities such as MRI or CT scans. This limitation
> restricts the applicability of the findings across the full spectrum of medical imaging technologies. The authors have listed
> the same as a limitation of the work.**
>
> At the time of this study, no CLIP-like VLSMs trained natively for 3D modalities exist. While it is true that there are essential imaging modalities such as
> MRI and CT scans, and studies on that will be interesting, the effort for developing models that work intrinsically for 3D modalities is outside the scope of
> this work. It is important to note that our framework, which includes prompt
> design and assessing how VLSM leverages language information, can still be
> helpful when 3D VLSMs are proposed in the literature and need evaluation.
> While we acknowledge that one could still use 2D slices of 3D images and test
> our method, we believe the diverse set of intrinsically 2D images, including
> ultrasound, X-rays, and photographic medical images, is already broad enough
> while 2D-specific to assess primarily 2D VLSMs. 3D modalities bring additional
> nuances when working as slices, such as how the 2D segmentation translates to
> 3D when looking at each slice independently and how to provide prompts for
> multiple slices. We will continue exploring 3D as a future work.
>
> > **Insights into modality-specific challenges and opportunities
> could guide future research towards modality-adaptive approaches.
> Discussion on how different imaging modalities affect the performance of VLSMs would be valuable information for the
> reader.**
>
> We agree that modality-specific challenges and opportunities could be very insightful in guiding future research. There are indeed some insights based on
> the results. The zero-shot segmentation performance was better for all the non-
> radiology datasets we used than for radiology datasets. The closeness of the non-radiology images and the external body parts (in ISIC and DFU datasets)
> with open domain datasets potentially leads to this.
> We will update the discussion in our paper to make these more straightforward. Furthermore, while there are potentially more modality-specific details
> that we have admittedly missed, generating such insights would necessitate further experiments, which are constrained by the scope of our work and the page
> limitations imposed by the conference.
>
> **References**
>
> [7] O. Ronneberger, P. Fischer, and T. Brox, “U-net: Convolutional networks
> for biomedical image segmentation,” in Medical Image Computing and Computer-
> Assisted Intervention–MICCAI 2015: 18th International Conference, Mu-
> nich, Germany, October 5-9, 2015, Proceedings, Part III 18, Springer, 2015,
> pp. 234–241.
>
> [8] Z. Zhou, M. M. Rahman Siddiquee, N. Tajbakhsh, and J. Liang, “Unet++:
> A nested u-net architecture for medical image segmentation,” in Deep
> Learning in Medical Image Analysis and Multimodal Learning for Clini-
> cal Decision Support: 4th International Workshop, DLMIA 2018, and 8th
> International Workshop, ML-CDS 2018, Held in Conjunction with MIC-
> CAI 2018, Granada, Spain, September 20, 2018, Proceedings 4, Springer,
> 2018, pp. 3–11.
>
> [9] L.-C. Chen, Y. Zhu, G. Papandreou, F. Schroff, and H. Adam, “Encoder-
> decoder with atrous separable convolution for semantic image segmen-
> tation,” in Proceedings of the European conference on computer vision
> (ECCV), 2018, pp. 801–818.

---

> > ### Author Response · Authors · 2024-03-18
> > **Official Comment by Authors (2/2)**
> >
> > > **I think it is important to briefly discuss any ethical considerations related to using VLSMs in medical image analysis,
> > including privacy concerns with patient data.**
> >
> > All the datasets we used were personally unidentifiable, made openly available
> > for research by original dataset creators, and commonly used by researchers.
> > The metadata available with the images does not disclose the identity of the
> > participant whose data was acquired. Since we exclusively use the images and
> > metadata provided with the datasets, VLSMs do not introduce any new ethical
> > challenges beyond those already existing in image-only medical image segmentation about privacy and anonymity.
> >
> > Moreover, as mentioned in *Appendix A: Impact on Society*, using VLSMs in
> > medical image segmentation contributes to fostering explainability and interoperability, particularly with humans in the loop, as language is more interpretable
> > than just images. This approach helps minimize ethical challenges associated
> > with black-box deep learning methods in medical applications.
> >
> > > **The motivation for segmentation in medical imaging is discussed in the paper; however, the discussion on the benefits
> > of using VLSM over image-only segmentation models has not
> > been included.**
> >
> > We want to emphasize the potential benefits of utilizing VLSMs over image-only
> > segmentation models, which motivated our research. As stated in the Introduction section, *VLSMs have the potential to allow for auxiliary information
> > incorporation through language prompts during segmentation. This approach
> > can enhance interpretability and robustness against domain shift and out-of-
> > distribution data.* Moreover, we conducted experiments to investigate whether
> > VLSMs outperform image-only models in adapting to distribution shifts. The
> > results of the experiments have been presented in the Results under the section
> > as VLSMs adapt better to distribution shifts. Our findings in the section state
> > that *VLSMs perform better in all the cases than the conventional models for
> > endoscopic datasets. The performance drops of VLSMs are smaller than that of
> > the conventional models when trained in a different distribution than that of the
> > test set.*

---

### Official Review · Reviewer_doLo · 2024-03-04

**Confidence:** 3
**Preliminary Rating:** 3
**Final Rating:** 3.5

**Summary:**

In this paper, the authors present a study exploring the performance, benefits and shortcomings of using vision-language segmentation models for medical-image segmentation. The authors curate a benchmark comprising 11 diverse medical image segmentation datasets, including different anatomies and image types. The authors then explore the performance of 2 different modality fusion techniques (aggregator vs language conditioning) on 2 different base models: CLIP and BiomedCLIP. Since the datasets are image-only, the authors explore several ways to add language prompts and explore the effect of different strategies on model performance. They find that in the medical domain as well, language conditioning in decoder (CRIS) performs better than using modality aggregation and conditioning (CLIPSeg). The authors show that the VLSM models are more robust to domain shifts than image-only counterparts but require more research to fully utilize the power of language conditioning.

**Strengths:**

* The paper is well written with crisp presentation and experiments are well designed to validate the hypothesis in testing. The authors also publish their datasets and code which is great for reproducibility and science.
* The benchmark is a valued contribution to the community to test new models/methods and easily compare results across different works.
* The paper presents negative results as well, leading to scope for further research and explorations to bridge the gap.
* I found the thorough experiments with prompts impressive and shed light on how different information is captured by the models and can help build better models for the task.

**Weaknesses:**

* The use of CLIP models for the study is suboptimal given there are many powerful open source VLM models available in the open source community. BiomedCLIP also has the disadvantage since no decoder is available and hence the authors train it from scratch which is not ideal with such a small dataset.
* Since prompt design is a core contribution of the paper, it would have been beneficial to include more details about it in the main paper rather than the appendix.

**Detailed Comments:**

1. What hyperparameters were considered when training these models? Since the comparison with several baselines is presented, it begs the question if the current performance of the VSLM models are due to suboptimal hyperparameters.
2. What is the motivation behind using CLIP as the base model for the paper when compared with other VLM models available in the open source community?
3. How was the decoder in BiomedCLIP variants trained? What was the data used? Did the model converge?

**Justification Of Final Rating:**

The authors addressed some of the concerns outlines in the review but hasn't provided a satisfactory argument for their experiment design and hence I recommend a borderline accept for the paper, but still believe the paper could benefit from a more robust experiment design. rather than a bag of many things.

**Justification Of The Preliminary Rating:**

Building on the previous work by Qin et. al. on studying the transfer behavior of vision language models for detection tasks, the current work extends the study to segmentation tasks in the medical domain. The experiments are well defined and the results reasonable but the paper lacks certain details about the experimental setup (as outlined in the review above) making it hard to judge the robustness of the results and conclusions.

**Questions To Address In The Rebuttal:**

1. What hyperparameters were considered when training these models? Since the comparison with several baselines is presented, it begs the question if the current performance of the VSLM models are due to suboptimal hyperparameters.
2. What is the motivation behind using CLIP as the base model for the paper when compared with other VLM models available in the open source community?
3. How was the decoder in BiomedCLIP variants trained? What was the data used? Did the model converge?

**Special Issue:**

No

---

> ### Author Response · Authors · 2024-03-18
>
> Thank you for your thorough review and constructive feedback on our work.
> We want to emphasize that
> our contribution in this paper goes beyond extending existing work on VLM-based object detection Qin et al. [1], which had demonstrated improved object
> detection performance with VLMs using automatically generated prompts. We
> propose novel ways to probe VLSMs and provide an evaluation framework to
> get insights into the nuanced roles of various attributes in language prompts
> and their value compared to image information during fine-tuning, which were
> not covered in Qin et al. [1]’s work.
>
> > **What hyperparameters were considered when training these models?
> Since the comparison with several baselines is presented, it begs the question of whether the current performance of the VSLM models is due to suboptimal hyperparameters.**
>
> We have reported the hyperparameters used in VLSMs in *Section 2.4: Implementation Details*.
> We provide additional information and the hyperparameters used in other baselines we implemented in *Appendix C: Experiment* of the supplementary material due to page limitations on the main body of the paper set by the conference guidelines.
> We conducted extensive grid searches and tuning within the limitations of our resources to get good enough hyperparameters that did not introduce vanishing and exploding gradients and performed optimally in most datasets.
> The details have been updated in *Appendix C: Experiment*.
> Furthermore, we ensured that all models converged well and provided meaningful results for comparison with baselines.
> Despite our efforts, we acknowledge the possibility that our hyperparameters may still be suboptimal, potentially resulting in slightly under-optimized models.
> We acknowledge that there is often a tradeoff when working with resource-intensive deep-learning models.
> However, we emphasize that we focused on understanding the nuances of fine-tuning VLSMs and language's role in that process.
> We present transparently the setup we worked on, provide a detailed appendix, and make the code open-source, allowing the community to assess its value and further build upon this work, for instance, when developing new methods to easily benchmark and analyze the role of the language prompts.
>
> > **Since prompt design is a core contribution of the paper, it
> would have been beneficial to include more details in the main
> paper rather than the appendix.**
>
> We have included the prompt design process in *Section 2.3: Generating Lan-
> guage Prompts.* We can revise the section to make it more straightforward. The
> details being moved to the Appendix were due to page limitations. Although
> we believe that the flow of the paper, experimental details, their results, and
> the main message we wanted to deliver are best managed within the page limit
> in the current version, we are open to suggestions and discussion to identify if
> any existing sections in the main body can be moved to Appendix to bring back
> the details on prompt design.
>
> **References**
>
> [1] Z. Qin, H. H. Yi, Q. Lao, and K. Li, “Medical image understanding with pre-
> trained vision language models: A comprehensive study,” in The Eleventh
> International Conference on Learning Representations, 2022.

---

> ### Author Response · Authors · 2024-03-18
> **Usage of CLIP-only models, and BiomedCLIP training**
>
> > **The use of CLIP models for the study is suboptimal, given that
> many powerful open-source VLM models are available in the
> open-source community.**
>
> > **What is the motivation behind using
> CLIP as the base model for the paper when compared with
> other VLM models available in the open-source community?**
>
> The scope of the study is to assess the performance of existing VLSM architectures trained on large-scale, open-domain image segmentation data with language prompts when fine-tuned for the medical domain. While many VLMs
> exist in the literature, most are only suitable for downstream tasks like image
> classification, image-to-text generation, and image retrieval. There are a limited
> number of VLSMs trained on large-scale image segmentation data with language
> prompts on top of an existing VLM and are mostly CLIP-based: CRIS [2],
> CLIPSeg [3], ZegCLIP [4], DenseCLIP [5]. To channel our efforts to better
> experimental designs on a wide range of datasets and prompts, we chose the
> scope of our study to be around two of the representative models: CRIS [2]
> and CLIPSeg [6]. The four VLSMs we created within the scope offered impressive diversity and representation—architectural variation in leveraging global
> level and token level information in prompts, trained end-to-end for referring
> image segmentation vs. fine-tuned only decoder with segmentation data, and
> VLMs pretrained on natural vs. medical domain. As with all benchmark and
> insight-generating studies like ours, there is always a tradeoff: implementing
> and experimenting with the exhaustive list of methods vs. picking representative and practical approaches and conducting insightful experiments that would
> be valuable to the community working on developing or using the techniques.
> We chose the latter option and believe we have provided a valuable and essential first step for building robust medical VLSMs by selecting many carefully
> collected datasets, designed prompts, and insightful experiments. With our
> source code being public in GitHub and the set of well-defined experiments and
> prompt generation, we emphasize our agreement with the reviewer’s comment
> that the *The benchmark is a valued contribution to the community to test new
> models/methods and easily compare results across different works.*
>
> > **BiomedCLIP also has a disadvantage since no decoder is available, and hence, the authors train it from scratch, which is not
> ideal with such a small dataset.**
>
> > **How was the decoder trained
> for BiomedCLIP variants? What was the data used? Did the
> model converge?**
>
> BiomedCLIP indeed has no trained decoder for segmentation. So, the decoders
> were trained entirely during our fine-tuning (BiomedCLIPSeg) or adapted from
> CLIPSeg’s pretrained model (BiomedCLIPSeg-D) and fine-tuned further. Experiments with these variants aimed to explore the tradeoff between two aspects: a. the familiarity of the BiomedCLIP encoder with the medical domain,
> b. the familiarity of CLIPSeg and CRIS models with the segmentation task and
> encoder-decoder interaction for the same, albeit in the natural domain. Given
> that the latter approach was observed to be more advantageous, our subsequent
> analysis only includes CLIPSeg and CRIS.
> The architecture of CLIPSeg’s decoder has been used for decoder training in
> BiomedCLIP variants. We trained this decoder in our fine-tuning experiments
> using the datasets present in our study until the model converged.
>
> **References**
>
> [2] Z. Wang, Y. Lu, Q. Li, et al., “Cris: Clip-driven referring image segmenta-
> tion,” in Proceedings of the IEEE/CVF conference on computer vision and
> pattern recognition, 2022, pp. 11 686–11 695.
>
> [3] L. H. Li, P. Zhang, H. Zhang, et al., “Grounded language-image pre-training,”
> in Proceedings of the IEEE/CVF Conference on Computer Vision and Pat-
> tern Recognition, 2022, pp. 10 965–10 975.
>
> [4] Z. Zhou, Y. Lei, B. Zhang, L. Liu, and Y. Liu, “Zegclip: Towards adapting
> clip for zero-shot semantic segmentation,” in Proceedings of the IEEE/CVF
> Conference on Computer Vision and Pattern Recognition, 2023, pp. 11 175–
> 11 185.
>
> [5] Y. Rao, W. Zhao, G. Chen, et al., “Denseclip: Language-guided dense pre-
> diction with context-aware prompting,” in Proceedings of the IEEE/CVF
> Conference on Computer Vision and Pattern Recognition, 2022, pp. 18 082–
> 18 091.
>
> [6] T. L ̈uddecke and A. Ecker, “Image segmentation using text and image
> prompts,” in Proceedings of the IEEE/CVF Conference on Computer Vi-
> sion and Pattern Recognition, 2022, pp. 7086–7096.

---

> > ### Comment · Reviewer_doLo · 2024-03-20
> >
> > I would like to thank the authors for addressing the questions in the review.
> >
> > I am curios as to why BiomedCLIP is included in the paper when
> > 1. It does not have a pretrained decoder and hence does not fit in the transfer learning scenario.
> > 2. Given the finetuning dataset is so small, no conclusions can be drawn for the model.

---

> > > ### Author Response · Authors · 2024-03-25
> > >
> > > BiomedCLIP indeed utilizes different fine-tuning settings compared to CLIPSeg and CRIS. However, BiomedCLIP was interesting to us, as it uses CLIP architecture but is pretrained using 15 million biomedical image-text pairs, which, although smaller than CLIP, is still a significant number. Therefore, we wanted to explore the following questions with BiomedCLIP, whose answers were uncertain before our experiments.
> > > - How does an encoder trained on large in-domain (medical images) perform compared to an encoder-decoder trained on large out-of-domain images?
> > > - How much do the representations from an encoder pre-trained in the same domain help the decoder adapt from scratch, even with a small dataset?

---

### Official Review · Reviewer_PqP8 · 2024-03-04

**Confidence:** 4
**Preliminary Rating:** 5
**Recommendation:** Oral
**Final Rating:** 5

**Summary:**

This paper investigates the extension of vision-language models (VLMs) to medical image segmentation (VLSM = vision-language segmentation model). The idea is to include text-based prompts along with the original input image allowing for a human-in-the-loop segmentation. 4 VLSMs derived from CLIP and BiomedCLIP are utilized against 11 datasets with a total of 10 possible prompts (with prompt 0 denoting the absence of a prompt). 3 of the 11 datasets are reserved to investigate the impact of domain shift on the model's behavior. Examples of prompts include prompts indicating the location of the foreground object such as "bottom right" or the size of the background object such as "small" as well as combinations of such concepts. Experiments are performed for both zero-shot and fine-tuning strategies.

**Strengths:**

=> Diversity of datasets in terms of there being both radiological as well as non-radiological datasets
=> Adding BiomedCLIP along with CLIP in order to compare between models trained on natural data as opposed to medical data
=> Identification that BiomedCLIP based models performs poorly compared to CLIP based models despite them being pre-trained on medical data
=> Showing visual examples of how models responds to prompts as well as examples of the model's behavior for incorrect prompts
=> Identifying that certain prompt types are more influential than others
=> Investigating the impact of prompts by substituting them with random words or semantically opposite terms
=> Detailed appendix

**Weaknesses:**

=> Most of the datasets 8/11 were non-radiological, and MRI and CT were completely excluded even though a 2D slice-by-slice approach allows for their (potential) inclusion
=> The impact of domain shift is only investigated for non-radiological datasets

**Detailed Comments:**

"a Dice score in the range of 20 − 70 for non-radiology datasets, with 67.98
being the highest Dice score for ISIC (Figure 2)." (Page 4) => Dice scores should either be given between 0 and 1 or a percentage sign should be added in front of the reported scores.

**Justification Of Final Rating:**

I am not completely convinced by the reasons given by the authors and am still curious as to how this will fare for standard radiological datasets such as MRI and CT, but overall I am satisfied with the paper.

**Justification Of The Preliminary Rating:**

Similar to vision transformers, prompting is another technique which has been borrowed from NLP, and has enjoyed relative success in computer vision. The extension of CLIP from classification to segmentation particularly in the medical domain is an interesting application allowing for the possibility of human intervention in the model's decision. This paper investigated this approach quite extensively highlighting both the advantages and potential shortcomings of such a methodology, and as such, should be welcomed as a valuable contribution to both medical image segmentation as well as prompting.

**Questions To Address In The Rebuttal:**

Why were most datasets non-radiological?

**Special Issue:**

Yes

---

> ### Author Response · Authors · 2024-03-18
>
> Thank you for your thoughtful review and for recognizing the importance of our work.
>
> > **Most of the datasets 8/11 were non-radiological, and MRI and CT were completely excluded even though a 2D slice-by-slice approach allows for their (potential) inclusion**
>
> While we acknowledge that we could use a 2D slice-by-slice approach on CT and MRI datasets to test our method, the slice-by-slice segmentation still needs to be translated back to 3D volume.
> The nuances regarding this, incorporating language have not been studied well and could introduce complications beyond the scope of this work.
> However, we acknowledge that investigating this aspect could be a valuable research direction for future work.
> Furthermore, we believe that our inclusion of a diverse set of intrinsically 2D images, with a fair balance of radiology as well as non-radiology modalities, provides a comprehensive assessment of primarily 2D VLSMs.
>
> > **The impact of domain shift is only investigated for non-radiological datasets**
>
> Among the datasets we utilized, the endoscopic datasets comprised multiple datasets focused on segmenting the same target, polyps, making them ideal for testing domain shift.
> However, for the radiology datasets selected, we could not find other datasets segmenting the same target organ.
> From a different perspective, other datasets should be similar since we utilize radiology reports and metadata to design prompts for radiology datasets.
>
> > **"a Dice score in the range of 20 - 70 for non-radiology datasets, with 67.98 being the highest Dice score for ISIC (Figure 2)." (Page 4)  Dice scores should either be given between 0 and 1, or a percentage sign should be added before the reported scores.**
>
> Thank you for the observation. We have updated the Figure and content accordingly.
>
> > **Why were most datasets non-radiological?**
>
> Of the 8 non-radiology datasets we used, six were endoscopic datasets for polyp segmentation.
> So, in a broader context, we chose three radiology and three non-radiology datasets, maintaining a fair balance.
> The selection of multiple endoscopic datasets allowed for more detailed analyses, such as cross-dataset evaluations to test robustness against distribution shifts.
> Likewise, it also facilitated the creation of a diverse, pooled dataset.
> Therefore, we extended the endoscopic datasets to six different datasets because it enabled the designing wider range of experiments.

---

### Author Response · Authors · 2024-03-18
**General response to everyone**

We thank all the reviewers for their valuable and thorough feedback.
It is encouraging that all three reviewers found this study valuable, providing essential insights through a comprehensive set of experiments covering many 2D medical image datasets of diverse modalities, organs, and pathologies.
We respond to the various points and reviewers' feedback in separate comments.

---

> ### Author Response · Authors · 2024-03-28
> **Changes made to the revised paper**
>
> We sincerely thank all the reviewers, as their comments and feedback allowed us to improve the paper. We have clarified and commented on most of the issues raised by the reviewers. Moreover, we have made minor changes to improve the paper's clarity as suggested by the reviewers, summarized below (also highlighted in red in the revised paper):
> - We have updated the result section to clarify the comparison against state-of-the-art image-only segmentation models to address Reviewer t1ep's concerns.
> - We have also updated the discussion to include modality-specific insights as suggested by Reviewer t1ep.
> - We have updated Appendix C of the supplementary section with details on the hyperparameter's search to address Reviewer doLo's concerns about the choice of hyperparameters.
> - We have updated the dice scores with percentage signs when not in the 0-1 range, as suggested by Reviewer PqP8.

---

### Comment · Area_Chair_XcDo · 2024-03-19
**The discussion period begins**

Dear reviewers and authors,

Thank you for your contribution to MIDL24. The discussion period begins! I encourage all reviewers and authors to participate in the discussion to address questions and clarify uncertainties.

Thank you!

---

### Meta-Review · Area_Chair_XcDo · 2024-04-03

**Recommendation:** Accept (Poster)
**Confidence:** 5

**Metareview:**

The reviewers agreed unanimously to accept this paper. I align with their decision. This paper provides a comprehensive evaluation study, which can offer valuable insights for peer researchers to continue exploring Vision-Language Models in medical imaging.

---

### Decision · Program_Chairs · 2024-04-05

Accept (Oral)